# Genome-Wide Identification of the MADS-Box Gene Family during Male and Female Flower Development in Chayote *(Sechium edule*)

**DOI:** 10.3390/ijms24076114

**Published:** 2023-03-24

**Authors:** Shaobo Cheng, Mingyue Jia, Lihong Su, Xuanxuan Liu, Qianwen Chu, Zhongqun He, Xiaoting Zhou, Wei Lu, Chengyao Jiang

**Affiliations:** College of Horticulture, Sichuan Agricultural University, Chengdu 611130, China

**Keywords:** Chayote, MADS-box, genome-wide, expression patterns, male and female flowers

## Abstract

The MADS-box gene plays an important role in plant growth and development. As an important vegetable of Cucurbitaceae, chayote has great edible and medicinal value. So far, there is little molecular research on chayote, and there are no reports on the MADS-box transcription factor of chayote. In this study, the MADS-box gene family of chayote was analyzed for the first time, and a total of 70 MADS-box genes were identified, including 14 type I and 56 type II MICK MADS genes. They were randomly distributed on 13 chromosomes except for chromosome 11. The light response element, hormone response element and abiotic stress response element were found in the promoter region of 70 MADS genes, indicating that the MADS gene can regulate the growth and development of chayote, resist abiotic stress, and participate in hormone response; GO and KEGG enrichment analysis also found that SeMADS genes were mainly enriched in biological regulation and signal regulation, which further proved the important role of MADS-box gene in plant growth and development. The results of collinearity showed that segmental duplication was the main driving force of MADS gene expansion in chayote. RNA-seq showed that the expression levels of *SeMADS06*, *SeMADS13*, *SeMADS26*, *SeMADS28*, *SeMADS36* and *SeMADS37* gradually increased with the growth of chayote, indicating that these genes may be related to the development of root tubers of ‘Tuershao’. The gene expression patterns showed that 12 SeMADS genes were specifically expressed in the male flower in ‘Tuershao’ and chayote. In addition, *SeMADS03* and *SeMADS52* may be involved in regulating the maturation of male flowers of ‘Tuershao’ and chayote. *SeMADS21* may be the crucial gene in the development stage of the female flower of ‘Tuershao’. This study laid a theoretical foundation for the further study of the function of the MADS gene in chayote in the future.

## 1. Introduction

MADS-box is a very important transcription factor widely existing in eukaryotes. MADS-box is an abbreviation of four genes initials: Mini chromosome maintenance 1 (*MCM1*) of *Saccharomyces cerevisiae* [1], AGAMOUS (*AG*) of *Arabidopsis thaliana* [2], DEFICIENS (*DEF*) of *Antirrhinum majus* [3], and the serum response factor (*SRF*) of *Homo sapiens* [4]. The N-terminal of the MADS-box protein contains a conserved domain of MADS-box (M) composed of about 58 amino acids [5]. According to the classification criteria of MADS gene structure and phylogenetic analysis, the MADS gene can be divided into type I and type II [6]. Type I includes Mα, Mβ and Mγ. The type II MADS gene is also called the MICK gene family because it contains four domains: MADS-box (M), intermediate region (I), keratin-like (K), and carboxyl-terminal (C). MICK type genes can be divided into MICK^c^ and MICK*, in which MICK^c^ includes 13 subfamilies [7]. The type I MADS gene structure contains only 1–2 exons and a highly conserved MADS-box (M) domain, but lacks a K domain, which is the characteristic of distinguishing type I and type II MADS genes. The type II MADS gene contains 6–8 exons and is also the most studied MADS gene [8].

The MADS-box gene plays an important role in plant growth and development, especially in floral organ development. The classical ABC model proposed that the development of the floral organ was controlled by different genes. In *Arabidopsis*, sepal development is controlled by class A genes (*AP1* and *AP2*), petal development is determined by class A and class B genes (*AP3* and *PI*) together, carpel development is determined by separate class C genes (*AG*), and stamen development is controlled by class B and class C genes together [9]. Subsequently, the ABC model was extended to the ABCDE model [9]. In this model, class D genes (*SKT*) and class C genes together control ovule development, class A genes and class E genes (*SEP1-4*) participate in sepal development, and class C genes and class E genes participate in carpel development. Stamen development is jointly regulated by class A genes and class C genes, and petal development is jointly regulated by A, B and E genes [10]. At present, in the widely accepted ABCDE model, except *AP2* in the class A gene, all other functional genes related to flower organ development are transcription factors, which contain a highly conserved DNA binding domain and belong to MADS-box family [11,12]. In addition, the MADS-box gene is also indispensable to regulate plant growth and development and resist various abiotic stresses [13]. Studies have shown that the MADS gene can regulate the development of sweet potato root tubers [14]. Overexpression of the *ZmSOC1* gene can improve maize yield and grain quality [15]. In addition, a large number of studies have shown that the MADS gene also plays an important role in response to abiotic stress. The overexpression of the *OsMADS25* gene in rice can improve salt tolerance and regulate auxin synthesis to promote root growth [16]. *AGL16*, a member of the *Arabidopsis* MADS-box family, negatively regulates stress response and is an important regulator to balance plant stress response and growth [17]. In addition, the MADS gene of *Rhododendron* also plays an important role under heat and waterlogging stresses [13].

Chayote (*Sechium edule*), 2n = 2x = 28, is a perennial herb climbing plant of Cucurbitaceae [18]. Chayote originated in Mexico and is widely planted in India, Brazil and Sichuan, Guizhou, Yunnan and other places in China. Its fruits, root tubers and leaves are rich in cucurbitoids, amino acids, vitamins, phenols and flavonoids, which not only make them widely used as an important vegetable, but also have the effect of treating diabetes and hypertension [19,20,21]. Chayote is mainly an edible fruit; and the chayote tuber is mainly an edible root. The fruit of chayote is developed from the expansion of female flowers after pollination. Its flowering period is from May to July and September to November every year. From May to July, only male flowers generally open. From September to November, the female flowers will bear fruit; the chayote tuber only blooms from September to November every year, and its female flower cannot normally expand into fruit [22]. Compared with Cucurbitaceae crops such as cucumber, melon, pumpkin and watermelon, the research on chayote is relatively sparse. The current research also focuses on the identification of fruit components. However, with the publication of the chayote genome, it provides great convenience for the research at the molecular level. Therefore, this study mainly uses the genome data of chayote to identify and analyze the MADS-box gene family and clarify the related mechanism of female and male floral organ development of chayote, and provide a theoretical basis for studying the molecular function of the MADS-box gene.

## 2. Results

### 2.1. Identification and Physicochemical Properties Analysis of the MADS-Box Gene Family of Sechium edule

A total of 70 MADS genes were identified from the genome of chayote and named *SeMADS01-70* based on the conserved MADS-box model (Table 1). The physical and chemical properties of these MADS genes were analyzed by ExPASY online software. The results showed that the 70 MADS-box genes were predicted to encode polypeptides from 63 (*SeMADS11*) to 674 (*SeMADS69*) amino acids, with a predicted molecular weight ranging from 7133.35 to 77846.80 Da. The isoelectric point of SeMADS ranged from 4.94 (*SeMADS64*) to 9.99 (*SeMADS48*), while the grand average of the hydropathicity of all MADS proteins except *SeMADS48* (0.069) was negative, ranging from −0.943 (*SeMADS02*) to −0.275 (*SeMADS14*), indicating that they were hydrophilic proteins. The total number of negatively charged residues (Asp + Glu) of SeMADS was 7 to 74, and the total number of positively charged residues (Arg + Lys) was 13 to 106. Subcellular localization predicted that all SeMADS were distributed in the nucleus.

### 2.2. Phylogenetic Analysis of SeMADS-Box

A phylogenetic tree was constructed using 70 SeMADS and 108 AtMADS proteins to clarify the evolutionary relationship of MADS in chayote. All MADS were divided into 15 clades marked with different colors (Figure 1). SeMADS, except for the *FLC*, formed 15 subfamilies. The maximum number of SeMADS found in the *AP1* branch was 11, followed by the *SVP* branch with 9 SeMADS, while only one SeMADS gene was found in the *AGL12* and *Bs* branches. The remaining SeMADS genes were distributed in 11 subfamilies, with two to eight genes in each subfamily.

### 2.3. Chromosomal Localization of SeMADS-BOX

Chromosome localization of 70 SeMADS-box genes was performed based on the genome annotation file of chayote (Figure 2). The results showed that 70 SeMADS genes were unevenly distributed on the other 13 chromosomes except the LG11 chromosome. The number of MADS genes on each chromosome ranged from two to ten. There were 10 MADS genes on chromosome LG09, accounting for 14.3%, and only two MADS genes on chromosome LG02, LG03 and LG12, accounting for 2.9%. There were eight genes on chromosome LG10, accounting for 11.4%, seven genes on chromosome LG01, LG05 and LG13, accounting for 10%, six genes on chromosome LG06, accounting for 8.6%, five genes on chromosome LG04, LG07 and LG14, accounting for 7.1%, and four genes on chromosome LG08, accounting for 5.7%.

### 2.4. Gene Structure and Conserved Motif Analysis of the SeMADS-Box

We used the GSDS program to analyze and display the exon and intron structures of 70 SeMADS genes to understand the diversity of the MADS-box gene structure of chayote (Figure 3A,B). The results showed that the structure of 70 SeMADS genes showed diversity, but the same subfamily showed similarity. The number of exons varies from 1 to 21, and *SeMADS69* has the largest number of 21 exons. Furthermore, we found that Mα, Mβ and Mγ mostly have no intron structure, or have one to two introns. These three subfamilies belonged to type I genes. This was the same as the type I gene of MADS in *Arabidopsis* and tomato, which may be due to the different trend of intron acquisition or loss or the reverse-transcribed origin of type I genes ancestors [23,24].

The protein sequences of 70 SeMADS genes were analyzed by the meme online program. A total of 10 motifs were obtained, named motifs 1–10 (Figure 3C). The results showed that all MADS proteins had motif 1 at the N-terminal of the sequence except for *SeMADS42*. This was a conserved motif that was thought to encode the MADS-box domain. We analyzed *SeMADS42* with SMART software and found that it has a MADS domain. In addition, motif 2 was distributed in other subfamilies besides Mα, Mβ and Mγ. Motif 2 encoded the K domain, which was a marker to distinguish type 1 and type 2 MADS genes. Motifs 3, 4, 5 and 6 were also widely distributed in type II MADS genes. Motifs 7 and 8 were only distributed in the Mβ subfamily. Motif 9 was distributed in the protein sequences of *SeMADS27*, *28* and *35*, and motifs 10 were only distributed in the *SOC1* subfamily.

### 2.5. Promoter Region Analysis of SeMADS-Box

Transcription factors can bind to the promoter region to regulate the expression of target genes. To predict cis-regulatory elements in the promoters of SeMADS, we analyzed 2 kb promoter sequences in the PlantCARE database. The results showed that all SeMADS genes contained light response elements (G-box, GT1 motif, Box4, ATCT motif, GATA motif, I-box, AE-box), which accounted for 51.7% of all cis-regulated elements. Hormonal response elements such as auxin response elements (TGA-element and AuxRR-core), abscisic acid response elements (ABRE), gibberellin response elements (TATC-box, p-box and GARE motif), MeJA-response (CGTCA-motif and TGACG-motif), salicylic acid response (TCA-element), and abiotic stress response elements such as drought induction (MBS), low-temperature response (LTR), anaerobic induction (ARE), defense and stress responses (TC-rich repeats) and wound-responsive element (WUN-motif) were widely distributed in MADS genes in various subfamilies. These response elements enabled chayote to cope with various adverse environmental changes and ensure its normal growth and development (Figure 4).

### 2.6. Collinearity Analyses of the SeMADS-Box within and between Species

The expansion of the gene family mainly includes five categories: whole-genome replication, scattered duplication, tandem duplication, segmental duplication, and proximal duplication [13]. We used BLASTp and MCScanX software (California, USA, http://chibba.pgml.uga.edu/mcscan2/, accessed on 12 January 2022) to analyze the replication type of SeMADS genes and conducted a collinearity analysis to explore the amplification mechanism of the chayote MADS gene family. The results showed that a total of eight SeMADS genes (11.43%) were found to form five tandem duplication gene pairs, and 50 SeMADS genes (71.43%) formed 38 segmental duplication gene pairs (Figure 5A and Appendix A). Therefore, segmental duplication was the main driving force for the expansion of the SeMADS genes family. It is worthy of note that most gene pairs were distributed in the same subfamily on the phylogenetic tree. We further calculated Ka/Ks to detect the selection pressure during the replication of 43 pairs of SeMADS genes (Appendix A). The results showed that only one of the tandem duplication gene pairs (*SeMADS45* and *SeMADS46*) had Ka/Ks > 1, indicating that it has been positively selected in the process of evolution. Other tandem duplication and all segmental duplication gene pairs had Ka/Ks < 1, suggesting that most SeMADS genes were under purification selection.

To further understand the evolution of the MADS gene family of chayote, we analyzed the collinear relationship between chayote and *Arabidopsis*, as well as some representative species of Cucurbitaceae (cucumber, melon, pumpkin and watermelon). The results showed that there were 55 pairs of homologous MADS genes between chayote and *Arabidopsis* (Figure 5B), 104 pairs of homologous genes with Cucurbitaceae pumpkin, followed by watermelon (84), cucumber (74) and melon (73) (Figure 5B, Appendix A). The number of collinear gene pairs between chayote and these species was not related to their own genome size. In addition, chayote and melon have no collinear gene pairs on chromosomes 1 and 5, while there were no gene pairs on chromosomes 5 and 11 between chayote and pumpkin (Figure 5B).

15 SeMADS genes (*SeMADS 12/16/19/20/30–31/33/37–38/40/47/49/58/63/65*) were not homologous with Arabidopsis, but were homologous with four species of Cucurbitaceae. In addition, we also found that *SeMADS23* was only homologous to melon and *SeMADS27*, and *SeMADS28* was only homologous to pumpkin. Interestingly, some SeMADS genes, such as *SeMADS07*, *SeMADS57* and *SeMADS67*, formed two to five collinear gene pairs, especially three to four collinear gene pairs with four species of Cucurbitaceae, suggesting that they play an important role in the evolution of the MADS gene family of chayote (Appendix A).

### 2.7. GO and KEGG Analysis of SeMADS-Box

To further understand the function of SeMADS genes, 70 SeMADS genes were analyzed by GO enrichment and KEGG pathway analysis. Seventy SeMADS genes were significantly enriched in 52 biological processes (BP), four molecular function (MF) and 3 cellular components (CC) (Figure 6A, Appendix A). In the BP category, three SeMADS genes were enriched in the regulation of flower development (GO:0009909) and the regulation of shoot system development (GO:0048831). The anchored component of plasma membrane (GO:0046658) and DNA binding transcription factor activity (GO:0000981) were the main enriched items in the CC and MF categories. The results of the KEGG pathway showed that 70 SeMADS genes were mainly significantly (Q < 0.05) enriched in MAPK signaling pathway-fly (ko04013), cGMP-PKG signaling pathway (ko04022), apelin signaling pathway (ko04371), and parathyroid hormone synthesis, secretion and action (ko04928) (Figure 6B, Appendix A).

### 2.8. Protein-Protein Interaction Network Analysis of the SeMADS-Box

Almost all life activities in cells depend on protein-protein interaction [25]. We predicted the protein interaction of SeMADS based on the homologous MADS gene in Arabidopsis using the STRING online database. As shown in Figure 7, *FUL* (ortholog of *SeMADS02*), *AP1* (ortholog of *SeMADS36*) and *AG* (ortholog of *SeMADS51*) could co-regulate the development of flower meristem. *SVP* (*SeMADS53* orthologous), *AGL24* (ortholog of *SeMADS10*) and *AGL6* (ortholog of *SeMADS09*) could participate in flowering regulation. *SHP1* (ortholog of *SeMADS01*) and *SHP2* (ortholog of *SeMADS22*) regulate ovule development. *AGL14* (ortholog of *SeMADS07*), AGL*21* (ortholog of *SeMADS52*) and *AGL44* (ortholog of *SeMADS54*) co-regulate root development.

### 2.9. Expression Profiles of the SeMADS-Box

The root tubers of chayote have important economic value. In order to further investigate the potential role of the MADS gene in root tubers, we conducted RNA-seq on the root tubers of ‘Tuershao’ at three different developmental stages. The results showed that most MADS genes were differentially expressed at different developmental stages (Figure 8). In the T1 period, the gene expression levels of *SeMADS01*, *SeMADS18*, *SeMADS27*, *SeMADS33*, *SeMADS55*, *SeMADS70*, *SeMADS08* etc. were relatively high. *SeMADS53*, *SeMADS15*, *SeMADS38*, *SeMADS01*, *SeMADS60*, *SeMADS58*, *SeMADS45* etc. genes were highly expressed in the T2 period. The expression levels of *SeMADS43* and *SeMADS57* were higher in the T3 period. It is worth noting that the expression levels of *SeMADS36*, *SeMADS37*, *SeMADS13*, *SeMADS28*, *SeMADS06* and *SeMADS26* gradually increased with the growth of ‘Tuershao’ time, indicating that these genes may be related to the root tuber development of ‘Tuershao’.

### 2.10. Expression Analysis of SeMADS-Box in Different Tissue and Stage

To investigate the chayote MADS-box genes expression of different tissues, we selected 12 MADS genes for quantitative analysis in roots, stems, leaves, tendrils, male flowers, and female flowers (Figure 9). Most MADS-box genes were highly expressed in the male flowers of ‘Tuershao’, including *SeMADS03*, *SeMADS12*, *SeMADS15*, *SeMADS19*, *SeMADS21*, *SeMADS42*, *SeMADS47*, *SeMADS49*, *SeMADS56*, and *SeMADS57*, while *SeMADS03*, *SeMADS15*, *SeMADS21*, *SeMADS42*, *SeMADS47*, *SeMADS49* and *SeMADS57* were highly expressed in the male flowers of chayote. *SeMADS12* and *SeMADS52* had the highest relative expression in the leaf of chayote and in root of ‘Tuershao’. *SeMADS54* was expressed in all organs in ‘Tuershao’ and chayote. These results indicated that SeMADS genes had tissue specificity.

To further verify the functions of SeMADS in the process of floral organ development, we analysed the 12 SeMADS genes expression from five developmental stages of male and female flowers in ‘Tuershao’ and chayote by qRT-PCR (Figure 10 and Figure 11). In the process of male flower development, *SeMADS03*, *SeMADS12*, *SeMADS15*, *SeMADS47* and *SeMADS52* were highly expressed at S4 in ‘Tuershao’, while *SeMADS42* and *SeMADS56* were expressed to a lower degree; the expression of *SeMADS03*, *SeMADS19*, *SeMADS52*, *SeMADS54* and *SeMADS57* was highest at S5 in chayote (Figure 10); it is speculated that these genes are related to the maturation of male flowers in ‘Tuershao’ and chayote. In the process of female flower development, most SeMADS genes showed a single peak trend in S2 or S3 (Figure 11), which indicated that these SeMADS genes were important to ‘Tuershao’ and chayote in the early and middle stage of female flower development. Across S1–S5, the expression of *SeMADS21* in the female flower of ‘Tuershao’ gradually increased (Figure 11). The trend of *SeMADS21* expression indicates the important role of *SeMADS21* in the female flower development of ‘Tuershao’.

## 3. Discussion

With the completion of genome sequencing, many MADS-box whole gene family analyses have been reported, such as *Arabidopsis* [23], tomato [24], and wheat [26]. In this study, we analyzed the genome of Chayote and found 70 MADS-box genes. Compared with other species of Cucurbitaceae, including watermelon (39) [27], cucumber (43) [28] and melon (62) [29], chayote has relatively more MADS-box genes. On the one hand, this phenomenon may be related to the genome size. The genome of chayote was 606.42 Mb, which was larger than those of these three plants [18]; On the other hand, during the evolution of Cucurbitaceae, 184 gene families in the genome of chayote contracted and 200 expanded, while 138 in watermelon contracted and 109 expanded, 193 in Cucumber contracted and 91 expanded, and 116 in melon contracted and 162 expanded [18]. More expansion genes in chayote may lead to more MADS-box genes [18]. In addition, the number of MADS-box genes in the Cucurbitaceae species was less than that in *Arabidopsis* (108) [23], wheat (201) [26] and tomato (131) [24]. It seems that Cucurbitaceae species have lost some MADS-box genes in evolution [27]. The MADS-box gene was usually divided into type I and type II. Type I included Mα, Mβ, and Mγ. Type II was further divided into MICK^c^ and MiCK*. According to the classification method of the MADS-box gene in *Arabidopsis*, the phylogenetic tree of the MADS gene family in Chayote was constructed. Seventy SeMADS genes were divided into 14 subfamilies, type I Mα (7), Mβ (3), Mγ (4); Type II MICK* (3) and MICK^c^ PI (3), *Bs* (1), *SVP* (9), *AGL12* (1), *AG* (6), *AGL15* (3), *AGL17* (5), *SOC1* (8), *AGL6* (2), *SEP* (4), and *AP1* (11). Some 80% of them belong to type II MADS, which is similar to that reported in the MADS gene family, indicating that type II MADS genes were more conservative than type I MADS genes [13,14,30]. The *FLC* gene was reported to regulate flowering with the spring flower pathway [14,27]. The *FLC* gene was not found in type II MICK^c^, indicating that chayote can blossom without going through the vernalization process, or the vernalization process was not inhibited by the *FLC* gene. The deletion of the *FLC* subfamily genes was also found in the watermelon and cucumber of Cucurbitaceae [27,28]. *BS* subfamily genes were involved in seed pigmentation and endothelial development [30]. We found a *Bs* gene *SeMADS49*, but no *Bs* gene was found in cucumbers, watermelon and melon in Cucurbitaceae [27,28,29]. *SeMADS49* was likely to be related to the process of seed and fruit development of chayote. In addition, we found an *AGL12* subfamily gene, *SeMADS55*. The study showed that *AGL12* was related to root development [31]. An *AGL12* gene was also identified in watermelon [27], indicating that the root development of chayote was likely to be regulated by *SeMADS55*.

Moreover, we noticed that 70 SeMADS genes were distributed on 13 chromosomes, except chromosome LG11. The number of MADS genes distributed on each chromosome was not positively correlated with chromosome length, and showed a random distribution on each chromosome. In addition, the physical and chemical properties of 70 SeMADS proteins showed that the subcellular localization of all proteins was in the nucleus, suggesting that the MADS gene of chayote may play a transcriptional regulatory role in the nucleus. Gene structure and conserved motif analysis showed that there were differences in the MADS gene of chayote. We found that the gene length of the type II gene was longer and contained more exons than that of the type I gene, which was also an important feature in distinguishing type I from type II. Different subfamilies contain different numbers and types of conserved motifs [30]. We found that motif 1 was the most conserved motif, which was related to the SRF domain. Motif 2, motif 3, motif 4 and motif 5 were mainly distributed in the type II MADS gene, which was related to the semi conserved k-domain and the low conserved I-domain and C-domain.

Gene replication plays an important role in the amplification and evolution of transcription factors. In this study, we found 43 MADS gene replication events, and 84% of SeMADS genes were generated through segmental replication, indicating that segmental replication plays an important role in the amplification of the chayote MADS gene. The same results were found in *Cyclocarya paliurus* [32], *Rhododendron hainanense Merr.* [13], and *Fagopyrum tataricum* [33]. In addition, we found that one of the five tandem replication gene pairs (*SeMADS45* and *SeMADS46*) underwent purifying selection by calculating Ka/Ks. This pair of genes belong to the *AP1* subfamily and have a similar gene structure, which is consistent with previous reports, suggesting that tandem replication genes may produce genes with similar functions after purifying selection [34].

Through collinearity analysis among different species, it was found that 55, 73, 74, 104 and 84 pairs of homologous gene pairs were identified between chayote and *Arabidopsis* and Cucurbitaceae plants (melon, cucumber, pumpkin and watermelon), suggesting that Cucurbitaceae species may have an expansion after divergence. Studies have shown that *Arabidopsis* and Cucurbitaceae plants have a WGD event of about 101–156 Mya [18], resulting in more homologous gene pairs in Cucurbitaceae plants than in *Arabidopsis*. In addition, the second WGD event occurred in Cucurbitaceae at 27–51Mya [18]. The phylogenetic tree analysis showed that pumpkin was relatively distant from melon, cucumber and watermelon, suggesting that there were more homologous gene pairs between chayote and pumpkin. The collinearity analysis of chayote, *Arabidopsis* and Cucurbitaceae plants (melon, cucumber, pumpkin and watermelon) showed that *SeMADS14/32/41–46/48/59* might be specifically expanded in chayote. *SeMADS05/12/16/19–20/25/30–31/33/37–38/40/47/49/58/61/63/65* was specifically expanded in Cucurbitaceae.

Cis-acting elements such as the light response element, hormone response element, and abiotic stress response element were found in the SeMADS genes promoter region. The research shows that chayote has strong growth ability and stress resistance [21]. These elements may be the key ones related to the growth, development and stress resistance of chayote fruit and root tubers. In addition, a GO and KEGG enrichment analysis also found that the SeMADS genes are mainly enriched in biological regulation, the MAPK signaling pathway—fly (ko04013), the cGMP PKG signaling pathway (ko04022), the Apelin signaling pathway (ko04371), and parathyroid hormone synthesis, secret and action (ko04928), further indicating that the SeMADS genes can not only regulate the growth and development of chayote, but also regulate the genes related to the active components of chayote. A protein interaction analysis showed that the interaction of SeMADS genes jointly affected the growth and development of chayote. It is worth noting that the gene expression of *SeMADS06*, *SeMADS13*, *SeMADS26*, *SeMADS28*, *SeMADS36* and *SeMADS37* gradually increased in three different periods of root tuber growth of ‘Tuershao’. They may be the key genes regulating the root tuber development of ‘Tuershao’.

Floral organ development is an important turning point in the process of plant growth and development [35]. The sequencing of the chayote genome has promoted the breeding values [18]. ‘Tuershao’ is a kind of chayote variety with a high yield of root tuber, but the female flower cannot expand normally to form fruit. The correct formation of the female/male flower could theoretically form normal fruit. MADS-box genes play an important role in floral organ development of angiosperms. In recent years, many studies have reported that the MADS-box gene is involved in the regulation of floral organ development. *LsMADS55* plays an important role in the development of floral organs of lettuce [36]. *JmMADS58* and *JmMADS29* play a positive regulatory role in the development of male and female flowers of *Juglans mandshurica* [35]. *AcANR1b* and *AcAGL11* are essential to the development of floral organs of pineapple, regulating the development of male and female flowers respectively [37]. *AcMADS04* is an important candidate gene for male and the female flower differentiation of kiwifruit [38]. In addition, more than 30 MADS-box genes were found to be highly expressed in flowers in *Dactylis glomerata*, indicating that they play a critical role in flower development [39]. However, the role of MADS-box gene in the floral organ development of chayote is still unclear. In this study, we selected 12 SeMADS genes for tissue specificity and flower development expression analysis. All SeMADS genes were highly expressed in the male or female flowers of ‘Tuershao’ and chayote. Most genes are highly expressed in male flowers at the S4 or S5 stage, which indicates that these genes may regulate the development of male flowers in ‘Tuershao’ and chayote. *SeMADS19* and *SeMADS54* are homologous with the *AGL15* and *SVP* genes of *Arabidopsis*. *AGL15* and *SVP* are flowering repressors in *Arabidopsis* [40], while in ‘Tuershao’ and chayote, *SeMADS19* and *SeMADS54* are highly expressed in male and female flowers, and the same phenomenon is also observed in *Cunninghamia lanceolata* [41], which suggests that similar inhibitors may exist in chayote to limit the development of male and female flowers. Furthermore, the differential expression of *SeMADS19* and *SeMADS54* genes in different stages of female and male flower development in ‘Tuershao’ and chayote may be the reason for their different flowering time. Further functional verification is necessary. The *SEP* gene of *Arabidopsis thaliana* is involved in the development of male flowers [23]. *SeMADS03* and *SeMADS47* are homologous genes of *SEP*, which are highly expressed in the male flowers of ‘Tuershao’ and chayote. *SeMADS03* and *SeMADS47* may be the crucial genes regulating the development of male flowers in the late stage of ‘Tuershao’ and chayote. *SeMADS21*, which is homologous with *PI*, is involved in the development of female flowers in *Arabidopsis* [23], and is highly expressed in the S1-S5 stage of female flower in ‘Tuershao’, suggesting that it may be the key gene in the regulation of the development of female flowers. These SeMADS genes could regulate the floral organ development of chayote, which needs to be further verified by systematic experiments.

## 4. Materials and Methods

### 4.1. Identification of MADS-Box Genes Family in Chayote

The genome data of chayote were obtained from the Cucurbitaceae website (http://cucurbitgenomics.org/, accessed on 12 January 2022). We then accessed the Pfam 35.0 online database on 12 January 2022 (http://pfam.xfam.org/, accessed on 12 January 2022) and downloaded the hidden Markov model (HMM) of SRF-TF (PF00319) and the K-box (PF01486) of the MADS gene family [33]. The MADS-box gene family was then identified based on the genome data of chayote using HMMER3.3.1 software (California, USA) (E-value of 10^−5^) (http://hmmer.org/download.html, accessed on 12 January 2022), SMART website (http://smart.embl-heidelberg.de/, accessed on 12 January 2022), which was used to further determine the screened MADS gene. Furthermore, the molecular weight (MW), isoelectric point (PI), total number of negatively charged residues (Asp + Glu), total number of positively charged residues (Arg + Lys), and grand average of hydropathicity (GRAVY) of MADS proteins were analyzed by the ExPASY online website (https://web.expasy.org/protparam/, accessed on 19 January 2022). The Plant-mPLoc website (http://www.csbio.sjtu.edu.cn/bioinf/plant-multi/, accessed on 19 January 2022) predicted the subcellular localization of the MADS protein.

### 4.2. Construction of MADS-Box Phylogenetic Tree

The MADS-box protein sequences of *Arabidopsis* were downloaded from the *Arabidopsis* database (https://www.arabidopsis.org/, accessed on 19 January 2022). There were multiple comparisons of MADS proteins in *Arabidopsis* and chayote using ClustalW. The phylogenetic tree was constructed by neighbor-joining (NJ) using MEGA-X software (version 7.0.26, California, USA). The specific parameters were as follows: Poisson model, same (homogeneous), pairwise deletion, and bootstrap method with 1000 replicates [42]. Based on *Arabidopsis* AtMADS genes, chayote MADS genes were divided into different subfamilies, and finally the phylogenetic tree was beautified with the iTOL tool (https://itol.embl.de/, accessed on 25 January 2022).

### 4.3. MADS-Box Family Genes Chromosomal Localization

Based on the annotation file of chayote, the chromosome length information was obtained. The 70 MADS genes of chayote were then located on different chromosomes using the MG2C online tool (http://mg2c.iask.in/mg2c_v2.1/, accessed on 25 January 2022) [32].

### 4.4. Gene Structure and Conserved Motif Analysis of MADS-Box

The gene structure of the MADS gene in chayote was analyzed by GSDS (http://gsds.gao-lab.org/, accessed on 27 January 2022), including UTL (untranslated region), CDS (coding sequence), and intron. Seventy MADS protein sequences were submitted to the MEME online tool (https://meme-suite.org/, accessed on 27 January 2022) to analyze the conserved motifs. The number of motifs was set to 10, the minimum width of motifs was set to 6, and the maximum was set to 50. Other parameters were set by default [43,44]. We then downloaded the result meme.xml file and used the visualize meme/mast motif pattern tool of TBtools software_0987663 (Guangzhou, China) to visualize the results [45].

### 4.5. Cis-Element Analysis of MADS-Box

The sequences of 2000bp upstream of the start codon of 70 chayote MADS genes were extracted, and the sequences were then submitted to the PlantCare online website (http://bioinformatics.psb.ugent.be/webtools/plantcare/html/, accessed on 27 January 2022) to query the cis-acting elements of each MADS gene [46]. We downloaded and simplified the PlantCare results, and finally visualized results using TBtools software_0987663 (Guangzhou, China) [45].

### 4.6. Collinearity Analyses of MADS-Box within and between Species

BLASTp and MCScanX software (California, USA, http://chibba.pgml.uga.edu/mcscan2/, accessed on 12 January 2022) were used to analyze the replication types of MADS-box genes in chayote [30], *Arabidopsis* and some representative species of Cucurbitaceae (cucumber, ChineseLong_v3; melon, DHL92; pumpkin, Cmoschata_v1; and watermelon, 97103_v2.5), and to analyze the collinearity within and between genomes. The genome data and annotation files of *Arabidopsis* and Cucurbitaceae were downloaded from the *Arabidopsis* database and the Cucurbitaceae website described above. The KaKs_Calculator2.0 was used to calculate Ka and Ks [32]. Finally, the Circos of TBtools was used to visualize the results [45].

### 4.7. GO and KEGG Analysis of MADS-Box

GO enrichment and KEGG pathway analysis were used to understand the function of MADS genes. The function of MADS genes were annotated using the eggNOG-mapper database (http://eggnog-mapper.embl.de/, accessed on 29 January 2022), and the results were then visualized using TBtools software_0987663 (Guangzhou, China) [45].

### 4.8. MADS-Box Protein Interaction Network Analysis

The protein-protein interaction network of MADS proteins homologous to *Arabidopsis* was used to further analyze the interaction relationship between MADS proteins of chayote. The protein-protein interaction network was displayed using the online website of STRING (https://string-db.org/, accessed on 29 January 2022).

### 4.9. MADS-Box Gene Expression Pattern

The expression data of ‘Tuershao’ root tubers of three different developmental stages (T1: early, T2: middle, and T3: mature stages) were obtained from our previous transcriptome data [22]. Briefly, the FPKM value of the MADS gene in transcriptome data was log2 standardized and displayed by heat map [47].

### 4.10. Plant Material, RNA Extraction and qRT-PCR Analysis

Chayote (*Sechium edule*) and ‘Tuershao’ (a high tuber-yield chayote cultivar) were cultivated in Chongzhou, Chengdu, China (30°63′ N, 103°67′ E). ‘Tuershao’ is a new variety selected by our research group many years ago. The female flower of ‘Tuershao’ cannot expand into fruit normally, as with the chayote. The edible root tuber in the underground is its main product, and the root tuber yield is much higher than that of chayote [22]. For the tissue-specific expression analysis, we collected the roots, stems, leaves, tendrils, and open male and female flowers of ‘Tuershao’ and chayote; for floral organ development expression analysis, the male and female flower samples of ‘Tuershao’ and chayote with different diameters were collected and named S1 to S5, then immediately frozen in liquid nitrogen and stored at −80 °C. Each sample contained three biological replicates. The total RNA of each sample was isolated by a Plant RNA Kit (OMEGA, Beijing, China), and the first strand cDNA was synthesized by a Script™ RT reagent Kit (TaKaRa, Beijing, China). The qRT-PCR was performed on a Bio-Rad CFX96 Real-Time System with 2X SYBR Green Abstart PCR Mix (Sangon Biotech, Beijing, China). The chayote actin gene was used as the internal reference and the relative expression levels of each MADS gene was calculated using the 2^−∆∆Ct^ method [22]. All of the primers used for qRT-PCR are shown in Appendix A.

## 5. Conclusions

In this study, we identified 70 MADS-box genes from the genome of chayote, including 14 type I and 56 type II MADS genes. A phylogenetic analysis showed that 70 SeMADS genes were divided into 14 subfamilies. There are differences in gene structure, conserved motifs and cis-acting elements among different subfamilies. The results of a collinearity analysis speculated that segmental duplication was the main reason for the MADS-gene expansion of chayote. A GO and KEGG enrichment analysis also showed that the MADS gene played a key role in the development of chayote. *SeMADS06*, *SeMADS13*, *SeMADS26*, *SeMADS28*, *SeMADS36* and *SeMADS37* may be the key genes regulating the root tubers development of ‘Tuershao’. *SeMADS03* and *SeMADS52* may play an important role in the maturation process of male flowers of ‘Tuershao’ and chayote. *SeMADS21* may be an important regulatory gene in the development stage of female flowers of ‘Tuershao’. In conclusion, our results provide a basis for the further study of the MADS gene in chayote.

## Figures and Tables

**Figure 1 ijms-24-06114-f001:**
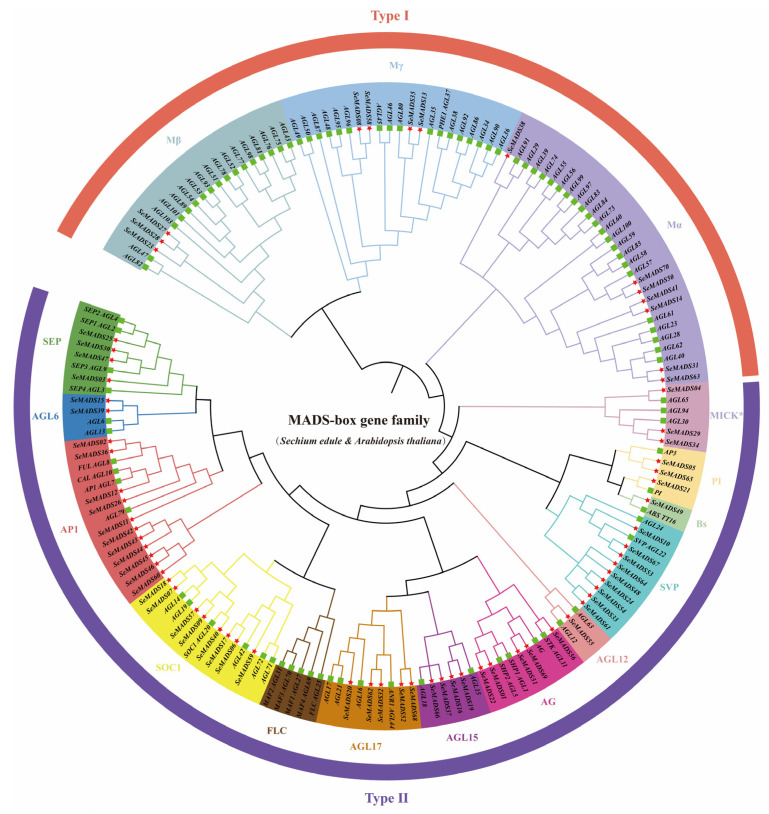
Phylogenetic tree based on the MADS-box proteins of *S. edule* and *Arabidopsis*. Different colors are used to distinguish fifteen subfamilies. The red star and the green square indicate *S. edule* and *Arabidopsis*, respectively.

**Figure 2 ijms-24-06114-f002:**
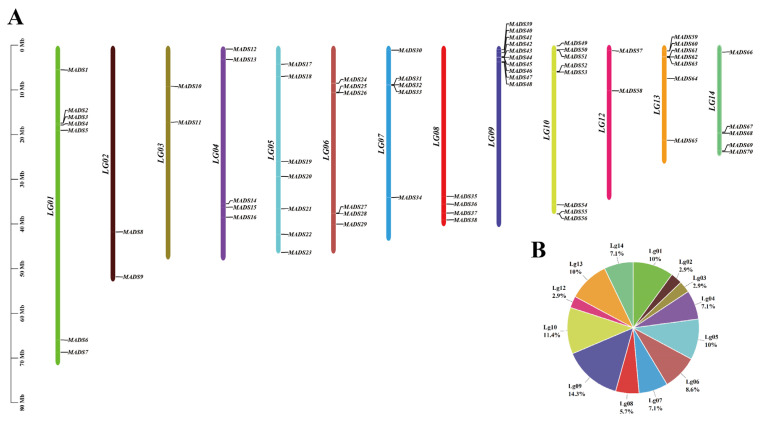
Chromosome location of the SeMADS-box. (**A**) Each chromosome is indicated in different colors. (**B**) The pie chart shows the percentage of MADS-box genes on each chromosome.

**Figure 3 ijms-24-06114-f003:**
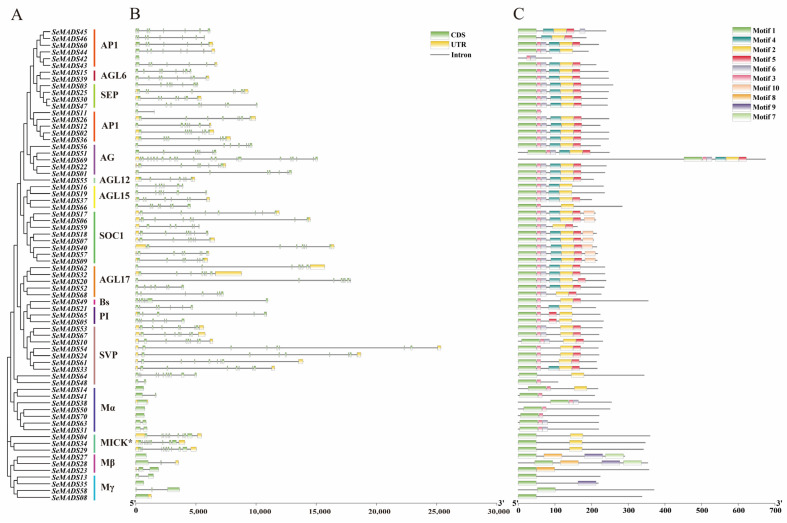
Protein conserved motif and gene structure analysis of SeMADS-box. (**A**) Neighbor-joining tree of SeMADS-box constructed by MEGA X. (**B**) Gene structure of SeMADS-box. (**C**) Protein conserved motif of SeMADS-box.

**Figure 4 ijms-24-06114-f004:**
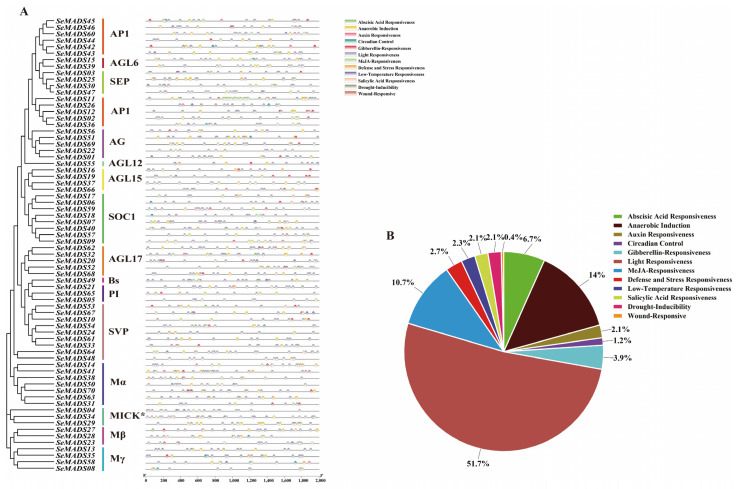
Cis-acting elements analysis of SeMADS-box in *S. edule*. (**A**) Distribution of cis-acting elements in each member of the SeMADS-box family. (**B**) Pie chart showing the percentage of different cis-regulatory elements predicted in SeMADS-box genes’ promoter sequences.

**Figure 5 ijms-24-06114-f005:**
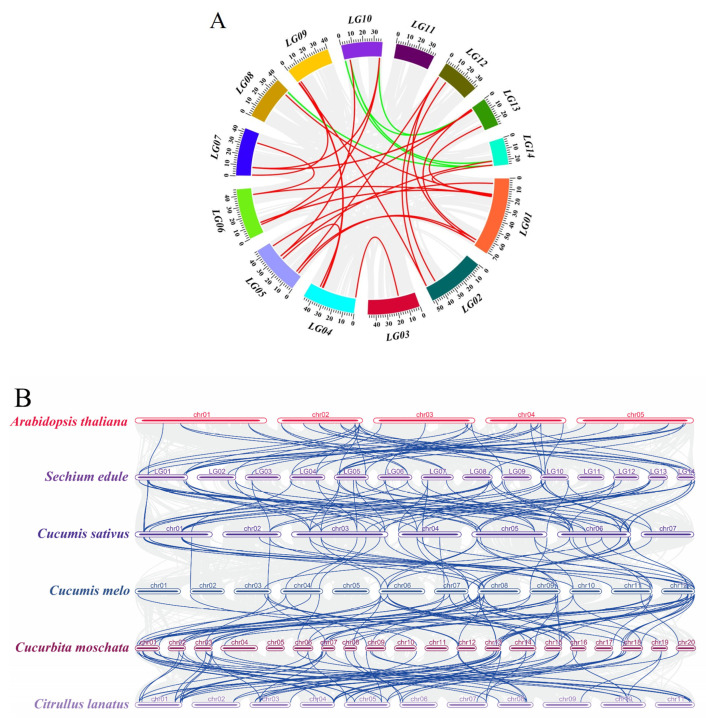
Collinearity analysis of MADS-box genes in *S. edule* and *Arabidopsis*. (**A**) Collinearity analysis of MADS-box genes in *S. edule*. The gray line represents all collinear regions in the genome of chayote, and the red line represents the collinear SeMADS-box in Se1-7; the green line represents the collinear SeMADS-box in Se8-14. (**B**) Synteny analysis of MADS-box genes between *S. edule* and *Arabidopsis* and species of Cucurbitaceae (cucumber, melon, pumpkin and watermelon). The blue line indicates SeMADS collinearity; the blue line indicates SeMADS-box collinearity.

**Figure 6 ijms-24-06114-f006:**
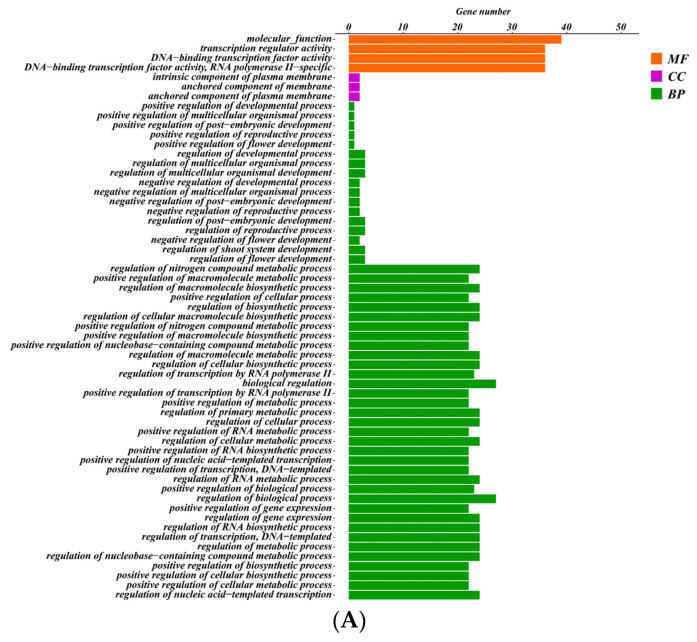
GO and KEGG pathway analysis of 70 MADS genes. (**A**) GO ontology of 70 MADS genes. The vertical axis represents different go terms, and the horizontal axis represents the number of genes enriched. MF: molecular function; CC: cellar component; BP: biological process. (**B**) KEGG pathway of 70 MADS genes.

**Figure 7 ijms-24-06114-f007:**
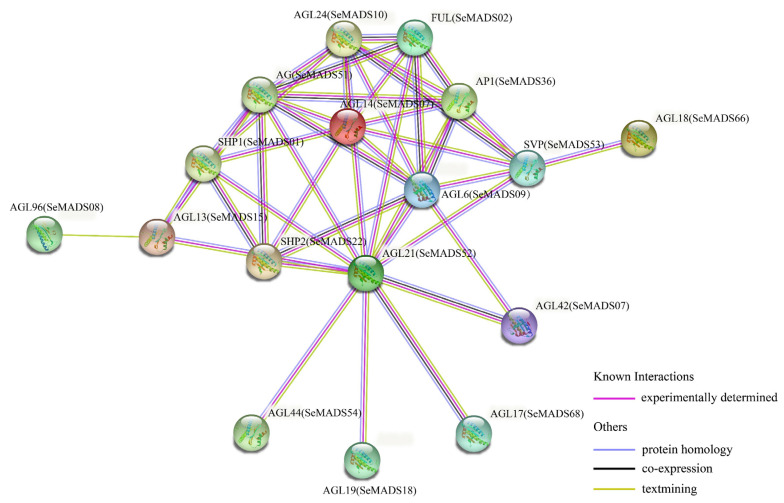
Functional interaction network of SeMADS proteins in *S. edule* based on their orthologs in *Arabidopsis*.

**Figure 8 ijms-24-06114-f008:**
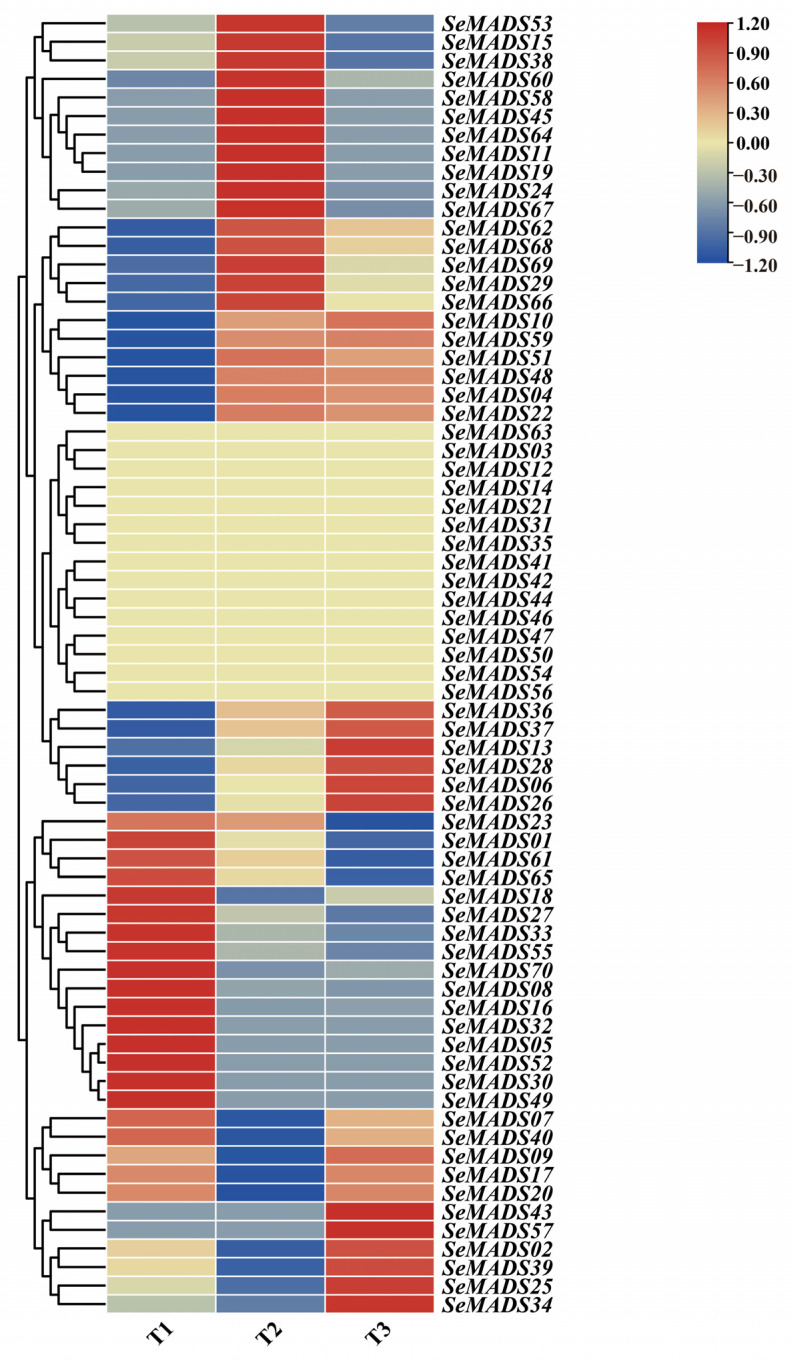
Expression of SeMADS genes in different developmental stages of root tuber in ‘Tuershao’. The heatmap represented the log2 (FPKM+1) value of SeMADS genes. T1: early stage, T2: middle stage, T3: mature stage.

**Figure 9 ijms-24-06114-f009:**
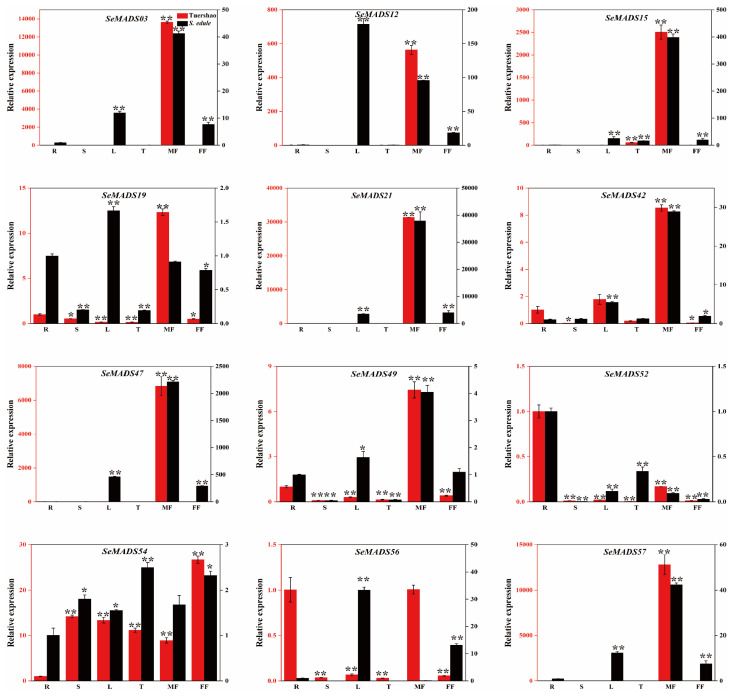
Expression of SeMADS-box genes of ‘Tuershao’ and *S. edule* in different tissues (R: Root, S: Stem, L: Leaf, T: tendril, MF: Male flower, FF: Female flower). Asterisks represent the significant difference at the same developmental stage point (* *p* < 0.05; ** *p* < 0.01).

**Figure 10 ijms-24-06114-f010:**
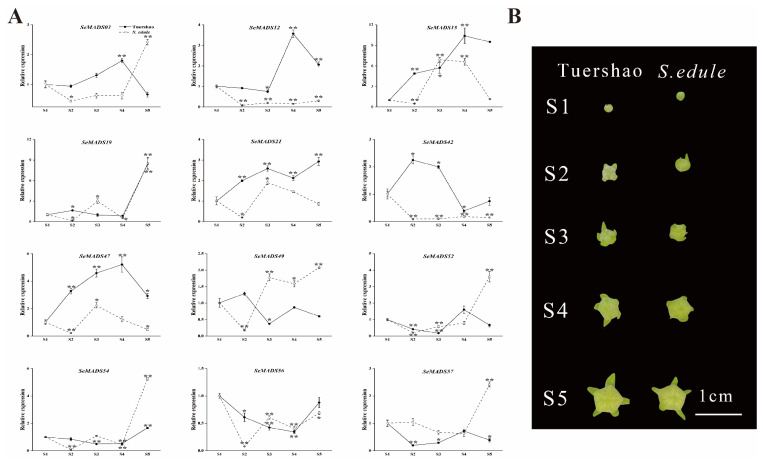
Comparison of expression patterns of the SeMADS-box at different male flower development stages in ‘Tuershao’ and *S. edule*. (**A**) SeMADS-box gene expression pattern. (**B**) Male flowers at different development stages. Asterisks represent the significant difference at the same developmental stage point (* *p* < 0.05; ** *p* < 0.01).

**Figure 11 ijms-24-06114-f011:**
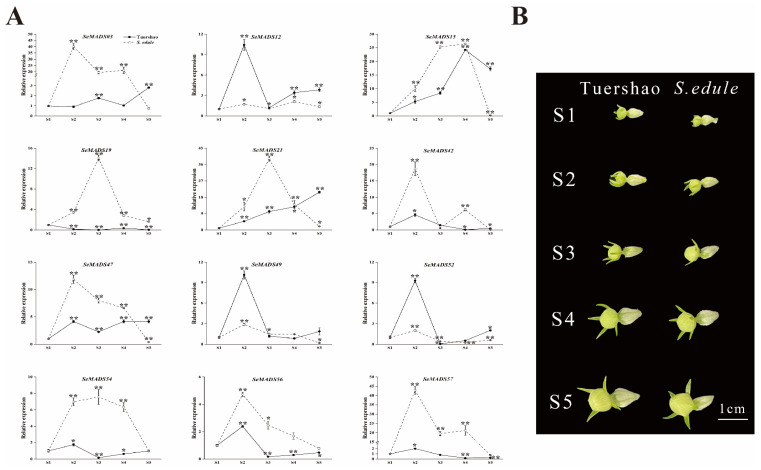
Comparison of expression patterns of the SeMADS-box at different female flower development stages in ‘Tuershao’ and *S. edule*. (**A**) SeMADS-box gene expression pattern. (**B**) Female flowers at different development stages. Asterisks represent the significant difference at the same developmental stage point (* *p* < 0.05; ** *p* < 0.01).

**Table 1 ijms-24-06114-t001:** Physical and chemical properties of MADS-box genes family in *S. edule*. (MW: molecular weight; PI: isoelectric point; ASP + Glu: total number of negatively charged residues (ASP + Glu); Arg + Lys: total number of positively charged residues (Arg + Lys); GRAVY: Grand average of hydropathicity; SLP: subcellular localization prediction).

Gene Name	Gene ID	Length(aa)	MW(Da)	PI	Asp + Glu	Arg + Lys	GRAVY	SLP
*SeMADS01*	Sed0027248	236	27,151.55	9.03	30	35	−0.822	Nucleus
*SeMADS02*	Sed0005514	247	28,710.73	9.66	30	42	−0.943	Nucleus
*SeMADS03*	Sed0021668	258	29,044.79	8.46	28	30	−0.599	Nucleus
*SeMADS04*	Sed0017917	359	40,697.70	5.72	44	36	−0.684	Nucleus
*SeMADS05*	Sed0023354	232	26,862.59	8.85	31	35	−0.730	Nucleus
*SeMADS06*	Sed0023514	211	23,885.32	8.81	29	34	−0.648	Nucleus
*SeMADS07*	Sed0028207	205	23,880.68	9.20	32	39	−0.726	Nucleus
*SeMADS08*	Sed0013720	337	37,643.72	5.73	46	38	−0.562	Nucleus
*SeMADS09*	Sed0009751	215	24,456.80	8.95	29	33	−0.705	Nucleus
*SeMADS10*	Sed0012619	230	25,797.58	8.36	33	35	−0.483	Nucleus
*SeMADS11*	Sed0018953	63	7133.35	9.92	7	13	−0.297	Nucleus
*SeMADS12*	Sed0008766	223	26,259.86	9.13	29	34	−0.850	Nucleus
*SeMADS13*	Sed0022320	223	25,657.81	9.40	26	34	−0.613	Nucleus
*SeMADS14*	Sed0018775	217	23,771.38	9.36	27	33	−0.275	Nucleus
*SeMADS15*	Sed0021213	245	28,232.24	8.61	30	34	−0.580	Nucleus
*SeMADS16*	Sed0006178	232	26,458.42	7.07	35	35	−0.559	Nucleus
*SeMADS17*	Sed0009958	210	23,801.38	9.28	29	37	−0.612	Nucleus
*SeMADS18*	Sed0019121	213	24,861.65	9.20	34	41	−0.800	Nucleus
*SeMADS19*	Sed0025694	235	26,337.13	5.94	35	30	−0.419	Nucleus
*SeMADS20*	Sed0016651	239	27,546.54	9.32	33	39	−0.699	Nucleus
*SeMADS21*	Sed0027083	211	24,902.45	8.74	33	36	−0.919	Nucleus
*SeMADS22*	Sed0014035	240	27,810.45	9.47	28	39	−0.870	Nucleus
*SeMADS23*	Sed0006669	356	41,143.66	7.14	41	41	−0.741	Nucleus
*SeMADS24*	Sed0011763	220	24,888.25	5.93	36	33	−0.677	Nucleus
*SeMADS25*	Sed0026823	246	28,172.01	8.97	29	33	−0.638	Nucleus
*SeMADS26*	Sed0012551	247	28,499.47	8.67	35	38	−0.754	Nucleus
*SeMADS27*	Sed0010395	290	33,536.67	5.33	50	35	−0.711	Nucleus
*SeMADS28*	Sed0020082	353	41,225.84	5.52	54	41	−0.471	Nucleus
*SeMADS29*	Sed0010468	341	38,775.45	6.27	40	36	−0.716	Nucleus
*SeMADS30*	Sed0006335	243	28,008.95	8.59	29	32	−0.645	Nucleus
*SeMADS31*	Sed0020653	219	24,716.41	9.59	20	29	−0.453	Nucleus
*SeMADS32*	Sed0015557	236	27,208.20	9.50	25	35	−0.695	Nucleus
*SeMADS33*	Sed0022681	215	24,209.69	9.34	28	34	−0.670	Nucleus
*SeMADS34*	Sed0013054	346	39,213.01	5.52	43	34	−0.696	Nucleus
*SeMADS35*	Sed0010970	218	24,726.63	8.86	27	31	−0.343	Nucleus
*SeMADS36*	Sed0023078	246	28,402.28	8.33	33	35	−0.782	Nucleus
*SeMADS37*	Sed0013411	200	22,993.30	9.95	25	37	−0.742	Nucleus
*SeMADS38*	Sed0017436	254	28,871.34	9.26	37	43	−0.518	Nucleus
*SeMADS39*	Sed0009092	246	28,305.01	8.93	29	34	−0.712	Nucleus
*SeMADS40*	Sed0003074	214	24,685.14	6.36	39	37	−0.797	Nucleus
*SeMADS41*	Sed0023327	208	23,157.95	9.44	24	32	−0.326	Nucleus
*SeMADS42*	Sed0017192	91	10,494.20	9.38	8	14	−0.305	Nucleus
*SeMADS43*	Sed0018117	212	24,447.88	9.38	31	39	−0.822	Nucleus
*SeMADS44*	Sed0024122	191	21,856.08	9.87	20	34	−0.665	Nucleus
*SeMADS45*	Sed0002596	239	28,379.56	8.50	31	34	−0.490	Nucleus
*SeMADS46*	Sed0012087	185	21,618.52	5.02	34	27	−0.864	Nucleus
*SeMADS47*	Sed0023127	243	27,905.73	8.60	29	32	−0.671	Nucleus
*SeMADS48*	Sed0004913	108	12,916.34	9.99	8	20	0.069	Nucleus
*SeMADS49*	Sed0000693	354	40,628.93	5.81	47	39	−0.671	Nucleus
*SeMADS50*	Sed0007340	250	28,333.50	6.31	32	28	−0.882	Nucleus
*SeMADS51*	Sed0021520	248	28,501.53	9.50	29	39	−0.751	Nucleus
*SeMADS52*	Sed0004053	234	26,720.89	9.22	30	36	−0.500	Nucleus
*SeMADS53*	Sed0027590	229	25,984.65	8.95	34	38	−0.650	Nucleus
*SeMADS54*	Sed0016270	218	24,221.75	9.21	30	35	−0.569	Nucleus
*SeMADS55*	Sed0005726	205	23,752.59	6.14	29	27	−0.412	Nucleus
*SeMADS56*	Sed0005214	224	25,715.39	9.35	27	35	−0.600	Nucleus
*SeMADS57*	Sed0019435	216	24,636.04	7.15	32	32	−0.664	Nucleus
*SeMADS58*	Sed0024714	370	42,194.47	8.49	42	45	−0.460	Nucleus
*SeMADS59*	Sed0024599	162	18,937.95	9.62	22	32	−0.665	Nucleus
*SeMADS60*	Sed0014292	219	25,201.49	5.78	35	32	−0.769	Nucleus
*SeMADS61*	Sed0025303	213	24,128.59	8.50	32	34	−0.645	Nucleus
*SeMADS62*	Sed0012885	236	27,235.31	9.28	30	37	−0.698	Nucleus
*SeMADS63*	Sed0009196	219	24,805.32	9.62	21	29	−0.623	Nucleus
*SeMADS64*	Sed0009551	343	38,508.87	4.94	41	28	−0.640	Nucleus
*SeMADS65*	Sed0007855	223	26,016.73	8.66	32	35	−0.674	Nucleus
*SeMADS66*	Sed0024257	283	32,502.48	5.35	48	38	−0.648	Nucleus
*SeMADS67*	Sed0017033	220	24,858.38	6.24	33	32	−0.585	Nucleus
*SeMADS68*	Sed0000564	226	25,924.68	8.34	30	32	−0.554	Nucleus
*SeMADS69*	Sed0004665	674	77,846.80	9.74	74	106	−0.466	Nucleus
*SeMADS70*	Sed0027724	220	24,635.77	7.78	28	29	−0.766	Nucleus

## Data Availability

Transcriptome sequencing data are available from the NCBI under project ID PRJNA842936.

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
