# Peer review of "Genome-Wide Identification of the MADS-Box Gene Family during Male and Female Flower Development in Chayote (Sechium edule)"

_ijms, 2023, doi:10.3390/ijms24076114_

Round 1
Reviewer 1 Report
In this manuscript, authors conducted comprehensive genome-wide identification of MADS-box genes in Sechium edule, and employed RNA-seq and qRT-PCR to analyze the expression pattern of seMADS-box genes in floral organ development.
Several issues in the manuscript are summarized as follows.
[1] In Line 95, the threshold value for HMMER should be added.
[2] In 2.6 section (Lines 134-135), the genomic version and genotype of cucumber, melon, pumpkin and watermelon (http://cucurbitgenomics.org/) used should be clearly stated. For example, there are the genomic sequences for three Cucumber genotypes or , namely ‘Chinese Long’, ‘Gy14’, and ‘PI183967’ on the web (http://cucurbitgenomics.org/). The v2 & v3 of ‘Chinese Long’, and the v1 & v2 of ‘Gy14’ are available on the web.
[3] In Line 138, ‘Ka and Ks’ was inaccurately written as ‘Ka and KS’.
[4] ‘SLP’ (subcellular localization prediction) was wrongly written as ‘SL’ in the table header and inconsistent with ‘SLP’ (subcellular localization prediction) in the table note in Table 1 (Line 190).
[5] What are the red star and green square at the tip of phylogenetic tree? Please state clearly in the figure legend (Lines 200-201).
[6] In Figure 5C, authors visualized the synteny relationship between S. edule and other four species of the Cucurbitaceae family on one-to-one mode. It might be better to visualize their synteny relationship on one-to-many mode, such as Fig.5B in Fu et al. (PMID: 33517348), and Figure 2 in Meng et al. (PMID: 36613699). JCVI (https://github.com/tanghaibao/jcvi) is a tool qualified for visualizing the one-to-many collinear relationship.
[7] ‘log2 (FPKM+1)’ was wrongly written as ‘log2 FPKM’ in the legend of Figure 8 (Line 347).
[8] The sentence (Lines 391-393) is not the result of the manuscript. Please add relevant reference.
Author Response
Dear Editors and Reviewers:
Thank you for your letter and for the reviewers’ comments concerning our manuscript entitled “Genome-Wide Identification of the MADS-Box Gene Family during male and female flower development in Chayote (Sechium edule)” (ID: ijms-2249203). Those comments are all valuable and very helpful for revising and improving our paper, as well as the important guiding significance to our researches. We have studied comments carefully and have made correction which we hope meet with approval. We used the revision mode of review to revise the paper. The main corrections in the paper and the responds to the reviewer’s comments are as flowing:
Responds to the reviewer’s comments:
Reviewer #1:
- In Line 95, the threshold value for HMMER should be added.
Answer: Thank you for underlining this deficiency. We have added the threshold value for HMMER in Line 96.
- In 2.6 section (Lines 134-135), the genomic version and genotype of cucumber, melon, pumpkin and watermelon (http://cucurbitgenomics.org/) used should be clearly stated. For example, there are the genomic sequences for three Cucumber genotypes or , namely ‘Chinese Long’, ‘Gy14’, and ‘PI183967’ on the web (http://cucurbitgenomics.org/). The v2 & v3 of ‘Chinese Long’, and the v1 & v2 of ‘Gy14’ are available on the web.
Answer: Thank you for the suggestion. We have added the genomic version and genotype in Line 135-136.
- In Line 138, ‘Ka and Ks’ was inaccurately written as ‘Ka and KS’.
Answer: We were really sorry for our careless mistakes. Thank you for your reminder. We have corrected the ‘Ka and KS’ into ‘Ka and Ks’.
- ‘SLP’ (subcellular localization prediction) was wrongly written as ‘SL’ in the table header and inconsistent with ‘SLP’ (subcellular localization prediction) in the table note in Table 1 (Line 190).
Answer: We sincerely thank the reviewer for careful reading. As suggested by the reviewer, we have corrected the ‘SL’ into ‘SLP’.
- What are the red star and green square at the tip of phylogenetic tree? Please state clearly in the figure legend (Lines 200-201).
Answer: Thank you for the suggestion. We have added relevant content to the figure legend in Line 202-203.
- In Figure 5C, authors visualized the synteny relationship between S. edule and other four species of the Cucurbitaceae family on one-to-one mode. It might be better to visualize their synteny relationship on one-to-many mode, such as Fig.5B in Fu et al. (PMID: 33517348), and Figure 2 in Meng et al. (PMID: 36613699). JCVI (https://github.com/tanghaibao/jcvi) is a tool qualified for visualizing the one-to-many collinear relationship.
Answer: We think this is an excellent suggestion. We have Re-visualized the synteny relationship between S. edule and other four species of the Cucurbitaceae family.
- ‘log2 (FPKM+1)’ was wrongly written as ‘log2 FPKM’ in the legend of Figure 8 (Line 347).
Answer: We feel sorry for our carelessness. In our resubmitted manuscript, the ‘log2 FPKM’ has been changed to ‘log2 (FPKM+1)’. Thanks for your correction.
- The sentence (Lines 391-393) is not the result of the manuscript. Please add relevant reference.
Answer: Thanks for your careful checks. We have added relevant reference.
We tried our best to improve the manuscript and made some changes under review mode in revised paper which will not influence the content and framework of the paper. We appreciate for Editors/Reviewers’ warm work earnestly, and hope the correction will meet with approval. Once again, thank you very much for your comments and suggestions.
Reviewer 2 Report
The authors have attempted to identify MADS-box gene family in Chayote's flower development. Overall, the manuscript is okay and seems like a standard methodology has been utilized. I have some comments that must be addressed.
Section 3.3. Chromosomal localization of SeMADS-Box"
This section can use some statistical analysis to justify the claims. What is the expected versus observed values on each chromosome?
Section 3.5: "Promoter region analysis of SeMADS-Box":
I am not sure what this section and the attached figures are meant to show without any statistical tests. The absence/presence of motifs means nothing.
Line 239: "To further study the regulatory network of SeMADS genes, we extracted..."
The analysis that follows is not a regulatory network analysis. What genes do these MADS target, or what targets them? I don't see any such info here.
line 301-304: "In the BP category, regulation of biological process (GO:0050789) and biological regulation (GO:0065007) were the main terms with 27 SeMADS genes. In addition..indicating that they may be involved in the regulation of gene expression."
These two terms are very broad terms that likely cover 80% of the genome. Frankly, it's trivial to even mention this.
Please report statistical tests for enrichments throughout the manuscript.
Author Response
Dear Editors and Reviewers:
Thank you for your letter and for the reviewers’ comments concerning our manuscript entitled “Genome-Wide Identification of the MADS-Box Gene Family during male and female flower development in Chayote (Sechium edule)” (ID: ijms-2249203). Those comments are all valuable and very helpful for revising and improving our paper, as well as the important guiding significance to our researches. We have studied comments carefully and have made correction which we hope meet with approval. We used the revision mode of review to revise the paper. The main corrections in the paper and the responds to the reviewer’s comments are as flowing:
Responds to the reviewer’s comments:
Reviewer #2:
- Section 3.3. Chromosomal localization of SeMADS-Box". This section can use some statistical analysis to justify the claims. What is the expected versus observed values on each chromosome?
Answer: We sincerely appreciate the valuable comments. We have re-visualized Figure 2 and added the percentage of SeMADS-box genes on each chromosome (Lines 209-217).
- Section 3.5: "Promoter region analysis of SeMADS-Box": I am not sure what this section and the attached figures are meant to show without any statistical tests. The absence/presence of motifs means nothing.
Answer: Thank you for the suggestion. We have added the percentage of each cis-acting element in Figure 4.
- Line 239: "To further study the regulatory network of SeMADS genes, we extracted...". The analysis that follows is not a regulatory network analysis. What genes do these MADS target, or what targets them? I don't see any such info here.
Answer: Thank you for the suggestion. We were really sorry for our wrong expression. We have made changes in lines 244 to 248.
- Line 301-304: "In the BP category, regulation of biological process (GO:0050789) and biological regulation (GO:0065007) were the main terms with 27 SeMADS genes. In addition, indicating that they may be involved in the regulation of gene expression." These two terms are very broad terms that likely cover 80% of the genome. Frankly, it's trivial to even mention this.
Answer: Thank you for the suggestion. We have deleted relevant contents.
- Please report statistical tests for enrichments throughout the manuscript
Answer: Thank you for the suggestion. We added statistical tests according to the reviewer's suggestion in Figures 9,10 and 11.
We tried our best to improve the manuscript and made some changes under review mode in revised paper which will not influence the content and framework of the paper. We appreciate for Editors/Reviewers’ warm work earnestly, and hope the correction will meet with approval. Once again, thank you very much for your comments and suggestions.
Round 2
Reviewer 1 Report
I have no comment.